# Seasonal Variation in Chemical Composition and Antioxidant and Antibacterial Activity of Essential Oil from *Cinnamomum cassia* Leaves

**DOI:** 10.3390/plants14010081

**Published:** 2024-12-30

**Authors:** Yanrong Cheng, Ying Fu, Dingze Gu, Yan Huang, Yongqi Lu, Yujie Liu, Xiulan Li, Xinyue Yao, Xinxuan Zhang, Wanying Jian, Peiwei Liu, Hong Wu, Yanqun Li

**Affiliations:** 1Guangdong Key Laboratory for Innovative Development and Utilization of Forest Plant Germplasm, South China Agricultural University, Guangzhou 510642, China; 20233154010@stu.scau.edu.cn (Y.C.); fuyingcassia@126.com (Y.F.); gudz@stu.scau.edu.cn (D.G.); 20223154028@stu.scau.edu.cn (Y.H.); lyq0721@stu.scau.edu.cn (Y.L.); yujieliu@stu.scau.edu.cn (Y.L.); iixiulan@stu.scau.edu.cn (X.L.); yaoxinyue@stu.scau.edu.cn (X.Y.); 202218330126@stu.scau.edu.cn (X.Z.); 202218330110@stu.scau.edu.cn (W.J.); 2Hainan Branch of the Institute of Medicinal Plant Development, Chinese Academy of Medical Sciences and Peking Union Medical College, Haikou 570311, China; brucelpw@aliyun.com; 3Medicinal Plants Research Center, South China Agricultural University, Guangzhou 510642, China

**Keywords:** *Cinnamomum cassia*, temperature, chemical variation, antibacterial effect, antioxidant activity

## Abstract

*Cinnamomum cassia* has been extensively utilized in traditional medicine systems worldwide. The essential oil (EO) content and composition are influenced by various external and internal factors, such as climate and harvest season, making it vital to determine the optimal harvest period for high-quality EO production. This study is the first to evaluate the chemical profiles, as well as the antioxidant and antibacterial activities, of *C. cassia* leaf oil across the four seasons. GC–MS and FTIR analyses revealed significant seasonal variations in the components. Spring and autumn leaf oils contained the highest EO (2.20% and 1.95%, respectively) and *trans*-cinnamaldehyde (92.59% and 91.10%, respectively). Temperature and humidity primarily affected EO and *trans*-cinnamaldehyde accumulation. *C. cassia* leaf oil demonstrated the strongest antibacterial activity, with a minimum inhibitory concentration (MIC) of 0.25 mg/mL against *S. aureus* and *L. monocytogenes* for the spring oil. The MICs for the other three seasonal samples were 0.5 mg/mL for *S. aureus*, *M. luteus*, and *L. monocytogenes*, and 1.0 mg/mL for *P. putida*. The minimum bactericidal concentration (MBC) of the EOs across all seasons against *S. aureus* ranged from 0.5 to 1 mg/mL. Winter leaf oil exhibited high antioxidant activity, primarily due to the presence of *cis*-cinnamaldehyde, caryophyllene, humulene, alloaromadendrene, γ-muurolene, *cis*-bisabolene, o-methoxycinnamaldehyde, and phenolics. This study provides essential data and valuable references for optimizing resource utilization and determining the ideal harvest time for *C. cassia* leaves.

## 1. Introduction

*Cinnamomum cassia*, an evergreen tree from the Lauraceae family, is a significant traditional Chinese herbal medicine. This species is prevalent in China, Vietnam, and many tropical regions [1]. With a rich history of medicinal use and cultivation in China, its primary production areas are Guangxi and Guangdong provinces in the south. It plays a vital role in international trade and is among the most commercially utilized spices [2]. Since ancient times, *C. cassia* has been valued as both a spice, due to its pleasant aroma and taste, and a critical crude drug in oriental medicine.

Cassia bark and leaf oils hold substantial economic importance, with *trans*-cinnamaldehyde being their major component. Cassia leaf oil is renowned for its antioxidant [3], antibacterial [4], and anti-inflammatory properties [5]. Commonly referred to as cassia oil, *C. cassia* leaf oil is used similarly to cassia bark oils in flavoring baked goods, confectionery, meat, sauces, pickles, soft drinks, and liqueurs. It is particularly significant in cola-type drinks [6]. The United Arab Emirates (UAE) is a major buyer of cassia leaf oil [2]. The demand for *C. cassia* remains strong, with promising prospects as consumption rises alongside population growth and the expansion of the soft drinks market.

Concerns over the safety of synthetic antioxidants have led to a focus on creating safe and effective alternatives. EOs extracted from natural plants, known for their excellent antioxidant activity, present a viable replacement for synthetic antioxidants in the modern food industry. Recent studies highlight the potential of natural products, including plant EOs, which exhibit strong antimicrobial activities and less bacterial resistance compared to artificial antibiotics [4]. Numerous studies have explored cinnamon’s antibacterial effects against food-borne bacteria [1,4,7]. *Staphylococcus aureus* and *Pseudomonas putida* are commonly linked to food spoilage and poisoning outbreaks. *Listeria monocytogenes* contamination leads to numerous food poisoning incidents annually, often associated with fish and fishery products, ready-to-eat salads, and various meat products [8].

Our previous research examined the differences in cassia bark oil components of various growth stages [9], species, and origins [10]. In a recent study, a comprehensive analysis was performed to compare the EO yields obtained from cinnamon/cassia leaves, revealing notable discrepancies in the chemical makeup of *C. cassia* and *Cinnamomum verum* leaf oils of various growth stages and species [11,12,13]. These studies have yielded valuable scientific data that can be utilized to evaluate and effectively utilize cinnamon resources. However, the season of harvest and weather conditions influence the content and composition of medicinal plants in the EOs [14]. It is well documented that the leaves of *Cistus ladanifer* [15], *Croton heliotropiifolius* [16], *Pistacia lentiscus* [17], and *Rosmarinus officinalis* [18] harvested in different seasons show significant differences not only in the chemical constituents of EOs but also in their biological activities. Consequently, investigating the seasonal variations in the composition and pharmacological activity of *C. cassia* leaves is important for improving their quality and economic benefits.

This study aimed to discern seasonal variations in the chemical composition, antibacterial activity, and antioxidant capacity of *C. cassia* leaves collected throughout the four seasons. GC–MS and FTIR spectra were employed to detect constituents and evaluate the quality of *C. cassia* leaf EO. Additionally, the antioxidant capacity and antibacterial activity of the EO were measured. The findings of this research can guide optimal harvest time for *C. cassia* in relevant industries and serve as a scientific reference for the development and utilization of *C. cassia* leaves.

## 2. Materials and Methods

### 2.1. Plant Materials

*C. cassia* leaves were harvested from 12-year-old trees cultivated in a forest in Zhaoqing, Guangdong Province, China, during April, July, and October 2019, as well as January 2020. The experimental site is situated at an altitude of 73.4 m above sea level, with geographical coordinates of 112°34′ E longitude and 23°36′ N latitude. The plants were identified by Rongjing Zhang of South China Agricultural University, Guangzhou, China.

### 2.2. Extraction Yield

The samples were placed in open containers in a dust-free environment to air-dry for 15 days. Sample preparation and EO extraction followed the method described by Li [11]. The air-dried leaves were ground into a powdered form (60 mesh) using a grinder. Thirty grams of fresh sample was extracted by hydrodistillation for 5 h, according to the Pharmacopoeia of the People’s Republic of China [19]. Methylene dichloride was used to extract volatile compounds from the water phase three times. The resulting yellow-colored volatile oils were weighed, and the collected volatile extract was stored at 4 °C for further analysis.

The EO yield percentage was calculated by dividing the weight of EOs by the weight of cinnamon leaf powder. These experiments were performed in triplicate. Data analysis was conducted using IBM SPSS Statistics 20, with statistical significance determined by analysis of variance (ANOVA). Results with a *p*-value of less than 0.05 were considered statistically significant.

### 2.3. GC–FID Analysis

The GC–FID analysis was conducted using San Agilent Technologies 7890 GC–FID coupled to a flame ionization detector and a capillary column of HP-5MS (30 m × 0.25 mm; 0.25-μm film thicknesses). The samples were diluted with dichloromethane to a concentration of 1 mg/mL, and subsequently, 1.0 μL of the diluted sample was injected for analysis. The injector temperature was set to 150 °C, while the detector temperature was set to 250 °C. The initial column temperature was maintained at 100 °C for 4 min and was gradually increased to 130 °C at a rate of 5 °C/min and held at 130 °C for 20 min. The linear velocity of the nitrogen carrier gas was 1.2 mL/min in a split ratio of 1:30.

To determine whether *trans*-cinnamaldehyde accumulation is influenced by different seasons, we employed the *trans*-cinnamaldehyde content as a quality evaluation standard. Calibration plots of the *trans*-cinnamaldehyde (CAS: 14371-10-9; TCI, Shanghai, China) standard were constructed from the peak areas (y) obtained for 5 different concentration solutions (x) to establish a standard curve (y = 22,201x + 2274.27, r² = 0.9994).

### 2.4. GC–MS Analysis

GC–MS analysis was conducted using an Agilent 7890A gas chromatograph (Santa Clara, CA, USA) coupled with a 5975C Plus mass spectrometer. Separation was achieved using a fused silica capillary column (Agilent Technology HP-5 ms, 5% phenyl methyl siloxane, 30 m × 0.25 mm i.d., film thickness 0.1 μm). The injector temperature was set to 150 °C and the detector temperature to 250 °C. Initially, the temperature was held at 100 °C for 4 min, then increased to 130 °C at a rate of 5 °C/min, and maintained at 130 °C for 20 min. Helium was employed as the carrier gas at a linear velocity of 1.2 mL/min with a split ratio of 30:1. Electron ionization (EI) was used as the ion source, with the ion source temperature set at 230 °C. Mass spectrometry scanning ranged from 30 to 550 amu with a scan time of 1 s. The samples were diluted to a concentration of 10 μg/mL using methylene dichloride, and subsequently, 1.0 μL of the diluted sample was injected for analysis.

Compound identification was validated by comparing the elution order with relative retention indices. Retention indices for all volatile constituents were determined using an n-alkane homologous series. Furthermore, EI mass spectra were compared with patterns in the NIST02 Mass Spectral Library. *Trans*-cinnamaldehyde was identified by co-elution with authentic standards.

### 2.5. FTIR Analysis

FTIR analysis was performed using a Nicolet 5700 spectrometer (Thermo Nicolet Corp. Madison, WI, USA) equipped with a deuterated triglycine sulfate (DTGS) detector. Infrared spectra were recorded over the range of 400–4000 cm^−1^ with a resolution of 4 cm^−1^. The ambient conditions were maintained at 25 °C with a relative humidity of 30%. Sample preparation involved accurately weighing 200 mg of dry KBr powder, which was used to produce two transparent KBr tablets having a diameter of 5 mm and a thickness of approximately 1 mm. EOs (2 μL) were applied onto the KBr tablets to create thin liquid films. Each sample underwent three measurements, with each measurement comprising three scans, totaling 9 spectra. The final sample spectrum was derived from the average of these scans. Background air spectra, water vapor, and CO_2_ interference were subtracted from the acquired spectra. Following this, baseline correction and curve smoothing were conducted using OMNIC 8.2 software. Subsequently, the spectral data were imported into Origin 9.0 and SPSS 20 software to standardize the inherent variations [20].

### 2.6. Ferric Reducing/Antioxidant Power (FRAP) Assay

The antioxidant capacity was assessed using commercial kits from Beyotime Institute of Biotechnology (Shanghai, China), which quantify the reduction of ferric ions. The *FRAP* assay was determined using the procedure reported in Li et al. [21]. In the reaction mixture, antioxidant agents reduce ferric-tripyridyltriazine (Fe^+3^ ± TPTZ), resulting in the formation of a blue complex termed Fe^+2^ ± TPTZ. The EO (10 μL, 10 mg/mL), extracts (10 μL, 10 mg/mL), and FeSO_4_·7H_2_O standard (10 μL, 0.15–1.5 mM) were dissolved in methanol and adjusted to 190 μL with the FRAP reagent. After incubation in the dark at 37 °C for 30 min, absorbance was measured at 593 nm. The concentration of Fe^+2^ ± TPTZ was determined by comparing the absorbance at 593 nm to the standard curve of Fe (II) solutions (ferrous sulfate heptahydrate).

### 2.7. ABTS Radical Scavenging Activity Assay

The total antioxidant capacity of the EO was evaluated using the ABTS method provided by Beyotime Institute of Biotechnology (Shanghai City, China) and was determined according to the technique reported by Li et al. [21] with some modifications. A solution containing 7.00 mM ABTS [2,2′-azino-bis(3-ethylbenzothiazoline-6-sulfonic acid)] and 2.45 mM potassium persulfate was incubated in darkness for 12 to 16 h. Subsequently, the solution was diluted with 70% ethanol until the absorbance reached 0.70 ± 0.02 at 734 nm. Next, the ABTS + solution was further diluted with 70% ethanol. The reaction mixture comprised EO (10 μL, 10 mg/mL) or Trolox standard (10 μL, 0.5–1.5 mM) and 200 μL of the ABTS solution. After a 10 min incubation in darkness at room temperature, absorbance was measured at 734 nm. Trolox, a water-soluble analog of vitamin E, was used as the reference standard for creating the calibration curve.

### 2.8. DPPH Radical Scavenging Assay

The DPPH radical scavenging activity was determined following a modified version of the procedure described by Lin [22]. Briefly, 0.2 mL of each EO sample, prepared at a concentration of 100 µg/mL in 80% methanol, was mixed with 1 mL of 0.2 mM ethanol DPPH solution. An ethanolic extract at the same concentration was also prepared. The reaction mixtures were vigorously shaken and then kept in the dark for 1 h. The reduction of DPPH radicals was measured by recording the absorbance at 517 nm against a blank without DPPH. Ethanol was used as the negative control, while gallic acid served as the positive control.

The percentage of DPPH inhibition was calculated using the following formula:$$\mathrm{Inhibition}\;\left(\%\right)=\left(\frac{Ab-Aa}{Ab}\right)\times 100\%$$

The percentage inhibition was calculated using the absorbance values of the DPPH radical with the plant extract sample (*Aa*) and the control (*Ab*). The inhibition percentage was plotted against the sample concentration to determine the 50% inhibitory concentration (IC50) from the regression equation. These values were then recalculated as the reciprocal of IC50 (1/IC50), referred to as antiradical power (ARP). A higher ARP value indicates greater antioxidant activity [23].

### 2.9. Total Phenolic Content (TPC)

Total phenolic content (TPC) was determined using the Folin–Ciocalteu reagent, following the method outlined by Abdelwahab [24], with tannic acid as the standard.

The extracted solution was filtered, and 0.1 mL of the ethanol extract was mixed with 0.6 mL of the Folin–Ciocalteu reagent, diluted 1:5 with distilled water. After thorough shaking, the mixture was allowed to react for 6 min. Subsequently, 1.25 mL of a 7% (*w*/*v*) sodium carbonate solution was added and shaken well. The mixture was then incubated for 1.5 h at room temperature, and absorbance was measured at 760 nm. The procedure was repeated for tannic acid standard solutions, and a standard curve was generated by plotting concentration (mg/mL) against absorbance (nm) (y = 0.0362x + 0.0744; R^2^ = 0.996). Here, y represents absorbance and x represents concentration in gallic acid (GAE) (n = 3). Total phenolic values were expressed as tannic acid equivalents (mg TAE/g DW).

### 2.10. Bacterial Strains

The reference bacterial strains utilized in this study were obtained from the Guangdong Microbial Culture Collection Center (GDMCC). The microorganisms included *S. aureus* (SA: ATCC29213), *P. putida* (PP: ATCC49128), *M. luteus* (ML: ATCC49732), and *L. monocytogenes* (LM: ATCC19115).

### 2.11. Agar-Disc Diffusion Method

A bacterial inoculum solution was prepared at concentrations of 4.00 mg/mL, 3.00 mg/mL, 2.00 mg/mL, 1.00 mg/mL, 0.50 mg/mL, 0.25 mg/mL, and 0.125 mg/mL and inoculated onto Mueller–Hinton agar plates. Sterile Whatman paper discs (6.00 mm in diameter) impregnated with 10.00 µL of each EO were placed on the surface of the inoculated agar. The inhibition diameters, including the disc diameter, were measured after incubation at 37 °C for 18–20 h. Gentamycin and oxytetracycline were used as positive controls.

### 2.12. Determination of Minimum Inhibitory Concentration (MIC)

MIC was determined using the broth macrodilution method as described by Zoubi et al. [25]. In this test, 8.00 mg of EO was diluted with 1.00 mL of dimethyl sulfoxide (DMSO) and mixed with 1.00 mL of Mueller–Hinton broth (MHB). Serial ½ dilutions were performed to achieve concentrations ranging from 4.00 mg/mL to 0.125 mg/mL. A 100 µL aliquot of bacterial suspension, prepared in saline solution to a final concentration of 10^8^ CFU/mL, was added to each tube. The tubes were incubated at 37 °C for 24 h. Bacterial growth was determined by adding 10 µL of triphenyl tetrazolium chloride (TTC) solution (5.00 mg/mL). The MIC value was identified as the lowest EO concentration that inhibited bacterial growth.

### 2.13. Determination of Minimum Bactericidal Concentration (MBC)

MBCs were determined by inoculating the negative control and samples on MHA plates. The MBC value is defined as the lowest concentration of EO that shows no bacterial growth after incubation at 37 °C for 24 h.

### 2.14. Statistical Analysis

Analysis of variance (ANOVA) and plotting were conducted using Origin 9.0 software. Statistical significance was considered at *p* < 0.05. Mean comparisons, principal component analysis (PCA), hierarchical clustering analysis (HCA), and Pearson correlation coefficients were performed using the SPSS 20 statistical package, version 16.0 for Windows (SPSS Inc., Chicago, IL, USA), to assess relationships among the phytochemical traits.

## 3. Results

### 3.1. Essential Oil Content and Composition at Different Seasons of Harvesting

The EO of *C. cassia* obtained by hydrodistillation appeared as a clear yellow liquid with a characteristic aroma. As depicted in Figure 1, the EO content of *C. cassia* varied across different seasons, ranging from 0.70% to 2.20%. The highest oil yield was observed in spring (2.20%), followed by autumn (1.95%). The EO content of leaves collected in winter (0.85%) and summer (0.70%) was lower. Notably, the EO yields in spring and autumn exceeded the minimum yield of 1.2% recommended by the Pharmacopoeia of the People’s Republic of China for *C. cassia* bark [19].

In this study, we used GC to determine the content of *trans*-cinnamaldehyde of *C*. *cassia* leaves, the main component of *C*. *cassia* EO. As shown in Figure 1b, the highest content of *trans*-cinnamaldehyde was recorded in spring and autumn, reaching 8.53 mg/L and 7.61 mg/L, respectively, while the lowest content was observed in winter, at 4.20 mg/L. To further clarify the chemical composition and content of *C*. *cassia* leaf EO in different seasons, the EOs extracted from *C*. *cassia* leaves were analyzed using GC–MS. As outlined in Table 1, 18 components were identified across all samples, representing 93.36–99.15% of the total oils. The results indicated that the four samples predominantly contained aldehydes, sesquiterpene hydrocarbons, and oxygenated sesquiterpenoids. The EO was characterized by a high percentage of aldehydes, which varied from 83.48% to 96.23% across different seasons. Aldehyde percentages in spring and autumn (96.23% and 93.54%, respectively) were significantly higher than in summer and winter (88.17% and 83.48%, respectively). The highest sesquiterpene hydrocarbon content (9.75%) was observed in winter. Oxygenated sesquiterpenoids ranged from 0.33% to 0.64% across the seasons.

Significant seasonal variations were observed in the content of individual components of EOs from *C. cassia* samples. In spring, the EO was dominated by *trans*-cinnamaldehyde (92.59%), followed by o-methoxycinnamaldehyde (3.4%) and dodecane (1.65%). In summer, the main constituents were *trans*-cinnamaldehyde (83.01%), o-methoxycinnamaldehyde (4.90%), and dodecane (1.00%). In autumn, *trans*-cinnamaldehyde (91.10%) was predominant, with dodecane (2.73%) and o-methoxycinnamaldehyde (2.08%) following. In winter, the dominant compound was *trans*-cinnamaldehyde (73.24%), accompanied by o-methoxycinnamaldehyde (8.07%), *cis*-bisabolene (4.36%), dodecane (1.87%), γ-muurolene (1.38%), humulene (1.28%), and alloaromadendrene (1.26%). The chemical composition of the EO varied significantly depending on the season of harvest. The results are consistent with the GC results. The *trans*-cinnamaldehyde content was highest in spring (92.59%) and autumn (91.10%), compared to summer (83.01%) and winter (73.24%). Seasonal variations also affected the minor components. In winter, the highest contents of o-methoxycinnamaldehyde (8.07%), *cis*-bisabolene (4.36%), alloaromadendrene (1.26%), γ-muurolene (1.38%), humulene (1.28%), and *cis*-cinnamaldehyde (1.08%) were observed. α-Muurolene was relatively high in summer at 1.16%. Notably, 2-methoxybenzaldehyde and γ-muurolene were absent in summer but present in other seasons.

### 3.2. Influence of Weather Parameters on the Essential Oil

The quality of EOs is often influenced by weather parameters. The data related to these parameters are presented in Figure 2a. Average temperature exhibited a significant positive correlation with *trans*-cinnamaldehyde, (Z)-3-phenylacrylaldehyde, and aldehydes (Figure 2b). Relative humidity also showed a significant positive correlation with *trans*-cinnamaldehyde and aldehydes, while exhibiting a significant negative correlation with α-cubebene, caryophyllene, and oxygenated sesquiterpenoids. Average precipitation demonstrated a significant positive correlation with *trans*-cinnamaldehyde and aldehydes and a significant negative correlation with α-cubebene, caryophyllene, and oxygenated sesquiterpenoids. Additionally, sunshine hours were significantly positively correlated with (Z)-3-phenylacrylaldehyde and oxygenated sesquiterpenoids. Among the weather parameters, average temperature and relative humidity had the greatest influence on *trans*-cinnamaldehyde.

The individual chemical constituents of the EO, observed at different harvest times across the four seasons, were also correlated among themselves (Table 2). *trans*-Cinnamaldehyde exhibited a significant negative correlation with alloaromadendrene. α-Cubebene displayed a significant negative correlation with caryophyllene. Additionally, the correlation between *trans*-cinnamaldehyde and (Z)-3-phenylacrylaldehyde was positive and significant.

### 3.3. FTIR Fingerprints of Leaves of C. cassia

The FTIR spectrum reveals an overall overlap of the characteristic absorption peaks of functional groups present in the chemical composition of the sample [26]. To further investigate the accumulation characteristics of chemical components in *C. cassia* leaf EOs, infrared fingerprint spectrum analysis was conducted on leaves harvested during the four seasons, as shown in Figure 3a. The infrared absorption spectrum of *C. cassia* leaves displays characteristic peaks in the wavelength range of 1800–400 cm^−1^. Several characteristic peaks were identified: (1) peaks at 1680 cm^−1^ and 1625 cm^−1^ correspond to the vibration stretching of the aldehyde carbonyl group (C=O) and C=C, indicating high levels of aldehydes and cinnamaldehyde in the EO; (2) the absorption band near 1450 cm^−1^ is attributed to the bending vibration absorption of C-OH from alcohol moieties; (3) the peak at 1124 cm^−1^ corresponds to the C–O stretching vibration and deformational vibration of C–OH groups; (4) the peak at 972 cm^−1^ is attributed to the bending vibration absorption of C-H groups; (5) the peak at 748 cm^−1^ corresponds to the =CH vibrational absorption of benzene rings; (6) the peak at 686 cm^−1^ represents alkene vibration absorption. To illustrate the overlap of the absorption spectrum in *C. cassia* leaves, the second derivative was used to examine the differences among the four samples (Figure 3b). The analysis shows that all the samples share common characteristic peaks, indicating similar chemical compositions. However, the winter sample displays strong absorption at the peaks of 1476, 1280, 950, and 513 cm^−1^, distinguishing it from the other three samples.

To observe the differences more intuitively, cluster analysis of the absorption features of each infrared spectrum was performed. Figure 4 illustrates that the samples were segregated into two primary clusters. Winter-harvested cassia leaves constituted one distinct cluster, distinctly different from those harvested in spring, summer, and autumn, indicating significant chemical composition differences. The small distance coefficients among the spring, summer, and autumn samples suggest that their leaf EO compositions are relatively similar.

### 3.4. Antioxidant Properties of EOs

This study evaluated the antioxidant activities of *C. cassia* leaf EOs harvested during the four seasons, employing the DPPH radical scavenging, FRAP, and ABTS assays, while using Trolox and gallic acid as reference standards. The DPPH and ABTS assays are based on radicals and the FRAP assay measures the reducing power of antioxidants by their ability to reduce Fe^3+^ to Fe^2+^. These radicals are commonly used to assess the free radical scavenging abilities of natural plants.

As shown in Table 3, the *C. cassia* EOs exhibited significant antioxidant activity. The DPPH, ARP, FRAP, and ABTS values for the leaf oils across the four seasons ranged from 15.87 to 44.46 μg/mL, 0.022 to 0.063, 17.49 to 35.94 mg Fe^2+^/g, and 7.15 to 18.05 mg Trolox/g, respectively. The winter leaf oils demonstrated the highest antioxidant activities, with an IC_50_ value of 15.87 μg/mL, ARP value of 0.063, FRAP value of 35.94 mg Fe ^2+^/g, and ABTS value of 18.05 mg Trolox/g. In contrast, the autumn leaf oils showed the lowest antioxidant activities, with an IC_50_ value of 44.46 μg/mL, ARP value of 0.22, FRAP value of 17.49 mg Fe ^2+^/g, and ABTS value of 7.15 mg Trolox/g. Phenolic compounds represent a prominent class of natural antioxidants. The total phenolic content ranged from 17.22 to 26.71 mg GAEs/g across different seasons. The results indicate a positive correlation between the antioxidant capacity of *C. cassia* leaf oils and their total phenolic content. For instance, the highest antioxidant activity in winter corresponded with the highest total phenolic content of 26.71 mg GAE/g, while the lowest antioxidant activity in autumn corresponded with the lowest total phenolic content of 17.22 mg GAE/g.

To explore the relationship between antioxidants and EO constituents, principal component analysis (PCA) was conducted to examine the correlation between 12 main bioactive substances and antioxidant indices in four samples of EOs (Figure 5). Two principal components (PC1: 67.80% and PC2: 4.84%) explained 92.64% of the total data variance, indicating that the antioxidant activity of *C. cassia* EOs was positively correlated with *cis*-cinnamaldehyde, humulene, alloaromadendrene, γ-muurolene, *cis*-bisabolene, and o-methoxycinnamaldehyde and negatively correlated with the other six components.

### 3.5. Antibacterial Activity of EOs

This study evaluated the antibacterial activity of *C. cassia* leaf EOs from different seasons against four bacterial strains: *S. aureus*, *M. luteus*, *L. monocytogenes,* and *P. putida* (Figure 6). The results showed significant differences in susceptibility among the strains. Increasing the concentration of the EOs enhanced the inhibitory activity against all tested bacteria. The EOs demonstrated superior antibacterial effects on *S. aureus* and *M. luteus*. Spring leaf EO exhibited the highest activity against *S. aureus*, with inhibition zones of 14.17 mm (++) at 25 mg/mL. At 100 mg/mL, *M. luteus* showed high sensitivity (+++) to the EOs from summer. The leaf EOs from all seasons outperformed the positive control (3.0 mg/mL) at certain concentrations. The EOs were less active against *L. monocytogenes* and *P. putida* compared to *S. aureus* and *M. luteus*. *L. monocytogenes* was moderately sensitive (++) to the EOs from spring, summer, and winter, while *P. putida* was moderately sensitive (++) to summer EO at 200 mg/mL. Autumn EO showed poor inhibitory effects on *L. monocytogenes*, and winter EO had weak effects on *P. putida*.

The *C. cassia* EOs exhibited inhibitory activity against all the tested microorganisms. In MIC determination, *S. aureus*, *L. monocytogenes*, and *M. luteus* were more susceptible, with values less than 1.00 mg/mL, compared to *P. putida*, which had a MIC of 1.00 mg/mL. *S. aureus* was the most sensitive strain, with a MIC of 0.25 mg/mL for spring EO, the lowest among all the seasons (Table 4). The MBC values were equal to or higher than the MIC for all tested strains. The MBC values of the spring, autumn, and winter EOs against *S. aureus* were 0.50 mg/mL, lower than those of the other samples (Table 4). Therefore, *C. cassia* leaf oil, particularly from spring, appears to be a promising antimicrobial agent.

## 4. Discussion

### 4.1. Influence of Harvest Season on Essential Oil Accumulation and Its Composition

Spatial and temporal variability significantly impacts the chemical characteristics of plants, making EO production highly dependent on environmental and pedoclimatic factors. Consequently, seasonal changes greatly affect plant secondary metabolites, altering both the yield and chemical composition of EOs [27], so from both medicinal material quality and economic perspectives, the correct harvest time can be very important. It has been reported that there was a higher oil yield in the rainy season (spring) of *Lippia alba* and *Ocimum gratissimum* leaf [28,29]. In this study, the average oil yield ranged from 0.70 to 2.20%, with the highest content in the warm rainy season (March). This oil content from spring was higher than that of leaves of 2-year branches with the highest oil content (2.12%) reported by Li et al. [10]. The annual variability of EO yield and chemical components is often related to temperature, sunshine duration, and rainfall, as well as attacks from fungal pathogens, especially during rainy months [30,31,32]. In this study, significant differences in oil content and *trans*-cinnamaldehyde were observed across different seasons, with temperature identified as the main factor influencing EO and *trans*-cinnamaldehyde accumulation. Higher oil production and greater contents of *trans*-cinnamaldehyde and aldehydes were recorded in spring and autumn, correlating with moderate temperatures (average monthly temperatures of 23.25 °C and 24.66 °C, respectively). In contrast, the lowest-temperature period (winter) showed a marked decrease in volatile production (0.70% oil content) and *trans*-cinnamaldehyde (73.24%). The results of hierarchical clustering analysis (HCA) based on FTIR data further indicated that the winter components were distinct from those of the other three seasons. These results suggest that moderate temperature ranges are more conducive to producing higher-quality oil. Climate conditions influence the biosynthesis of secondary metabolites by affecting the activity of biosynthetic enzymes in plants [33]. Each compound’s biosynthesis requires appropriate precursors, enzymatic activity, and climatic conditions (e.g., temperature and photoperiod) available at specific phenological growth stages. Consequently, variations in EO quality throughout the plant growth season are part of the plant’s natural mechanism [34]. In this study, climatic conditions such as temperature significantly affected aldehyde synthesis by influencing the activity of aldehyde biosynthetic enzymes. This, in turn, impacted the aldehyde biosynthesis rate, decomposition rate, and volatilization rate, ultimately determining the seasonal variation in EO yield and components in the leaves of *C. cassia*. Additionally, relative humidity exhibited a significant positive relationship with *trans*-cinnamaldehyde, indicating that higher humidity also promotes the accumulation of EOs and *trans*-cinnamaldehyde. Variations in oil composition may also be associated with fungal pathogen attacks, particularly during rainy months [32]. Recently, Ye et al. [35] reported that cinnamyl-CoA reductase is involved in *trans*-cinnamaldehyde biosynthesis in *C. cassia*. Understanding how environmental factors such as temperature influence *trans*-cinnamaldehyde accumulation and applying this knowledge to production holds significant potential for various industries.

Temperature affects the ratio of different metabolites, with compounds like o-methoxycinnamaldehyde, alloaromadendrene, α-cubebene, caryophyllene, humulene, γ-muurolene, *cis*-bisabolene, sesquiterpenoids, and oxygenated sesquiterpenoids being more abundant in both colder winters and hotter summers. Cold stress, a major limiting factor for plant productivity and distribution, particularly impacts terpenoids [36]. It can regulate rate-limiting enzymes such as phenylalanine ammonia-lyase (PAL) and 3-hydroxy-3-methylglutaryl-CoA reductase (HMGR) in the terpenoid biosynthesis pathway, significantly promoting terpenoid accumulation [36].

This study shows that the EO content and composition of *C. cassia* leaves are significantly influenced by seasonal and environmental factors. Leaves harvested in spring and autumn are particularly suitable for extracting high-quality volatile oil.

### 4.2. Influence of Harvest Season on Antioxidant Activity

*C. cassia* EOs demonstrate strong antioxidant capability [37,38]. Factors influencing the chemical composition of plant EOs, such as genetic factors, environment, growth stage, and physiological factors, also affect the antioxidant properties of plants [39]. The highest antioxidant activity in winter leaf oil, which contains high levels of phenolic compounds and specific terpene derivatives, implies a significant antioxidant capacity. Phenolic compounds are key determinants of a plant’s antioxidant activity, and the total phenol content of *C. cassia* leaves varies with the season. The highest total phenolic content was observed in leaves harvested in winter. This seasonal variation in phenolic content may be linked to biochemical changes and different stress responses at various growth stages [21]. Several studies have reported a positive correlation between lower temperatures and higher phenolic compound content [40,41]. The lower winter temperatures likely contributed to the increased phenolic content compared to other seasons. These outcomes align with Farhadi [42], who reported phenolic compound accumulation in *Achillea millefolium* at different growth stages. The antioxidant capacities of *C. cassia* leaf oil from different seasons showed a significant relationship with phenolic content. The highest antioxidant activity in winter leaf oil is likely due to its high phenolic content (Table 4). This result is consistent with studies indicating that phenolic compounds significantly contribute to the antioxidant capacities of *C. loureirii* [43]. High phenolic content enhances the rate of hydrogen transfer to free radicals and increases inhibitory strength [41].

Antioxidant activity was positively correlated with changes in humulene, alloaromadendrene, γ-muurolene, and *cis*-bisabolene in *C. cassia* leaf oils. These compounds, which belong to monoterpenoids and sesquiterpenes, have demonstrated strong antioxidant activity [1,44]. Belahcene [45] reported that the antioxidant activity evaluated through radical scavenging assays might be attributed to the high content of specific terpenes with conjugated double bonds and oxygenated monoterpenes. Thus, the antioxidant activity of EOs may be associated with the presence of monoterpenoids and sesquiterpenes [46]. Li [43] also reported that the antioxidant activity of *C. loureirii* bark oils positively correlated with γ-muurolene and negatively correlated with *trans*-cinnamaldehyde. While phenolic compounds are significant contributors to the antioxidant activity of C. cassia EO, other components, such as terpene derivatives, may also play a role in the overall antioxidant capacity of the oil. In conclusion, *C. cassia* leaf oil, especially when harvested in winter, can serve as a high-quality raw material for natural antioxidant products and pharmaceutical applications.

### 4.3. Influence of Harvest Season on Antibacterial Activity

Plant EOs exhibit a broad spectrum of inhibitory activity against various Gram-positive and Gram-negative bacterial pathogens, although the interaction between oils and different bacteria can influence the antibacterial effect [47]. This study investigated the antimicrobial activity of *C. cassia* leaf EO at different concentrations and seasons against *S. aureus*, *M. luteus*, *L. monocytogene*, and *P. putid*. The results demonstrated that EO from *C. cassia* leaves harvested in spring exhibited stronger bacteriostatic effects than those harvested in other seasons, likely due to the higher levels of *trans*-cinnamaldehyde in spring. EOs containing aromatic phenols or aldehydes have significant antibacterial activity, whereas those containing terpene ethers, ketones, or oxides exhibit weaker activity [48,49]. Isolated cinnamaldehyde effectively inhibits the growth of a range of microorganisms, including bacteria, molds, and yeasts, and suppresses toxin production by these microorganisms [50]. The antibacterial mechanisms of cinnamaldehyde include targeting the bacterial cell wall, biofilm formation, quorum sensing system, cell metabolism, and survival factors, thereby enhancing antibiotic efficacy and overcoming resistance [51,52].

*P. putida* proved more resistant to the oils, persisting even at high concentrations [53]. This study is among the first to report the antibacterial activity of *C. cassia* leaf EO against *L. monocytogenes* and *P. putida*. The results indicated that *C. cassia* leaf oil exhibits good antibacterial activity against these pathogens. At a concentration of 200 mg/mL, leaf EOs from various seasons showed moderate sensitivity against *P. putida* and *L. monocytogenes*, with inhibition zones above 10 mm.

These bacteria are widespread, can survive extreme conditions, and cause severe diseases in humans and animals [54]. Controlling these bacteria is vital for food safety and public health. *C. cassia* leaf oils, especially those collected in spring, show significant potential as food preservatives and sanitizing agents, highlighting their promise as new antimicrobial and protective agents in the food industry.

## 5. Conclusions

To the best of our knowledge, this study is the first to analyze the chemical composition, antioxidant activity, and antibacterial activity of *C. cassia* EO across different seasons. The EO content and composition fluctuate with the time of year, with temperature and humidity likely being the primary factors influencing final oil yields and *trans*-cinnamaldehyde content. Currently, the utilization rate of *C. cassia* leaves is low, and the EO derived from the leaves has a milder taste compared to bark oil, presenting substantial potential for exploitation. Spring and autumn yielded the highest EO and *trans*-cinnamaldehyde levels, making these seasons optimal for harvesting high-yield, high-quality *C. cassia* leaf EOs.

Significant differences in antioxidant capacity were observed among EOs from different seasons. The winter leaf oil exhibited the highest antioxidant capacity, potentially due to the presence of compounds such as caryophyllene, humulene, alloaromadendrene, γ-muurolene, *cis*-bisabolene, and o-methoxycinnamaldehyde. *C. cassia* leaf oil demonstrated strong antimicrobial activity against *S. aureus*, *M. luteus*, *L. monocytogenes*, and *P. putida*, with spring leaf EO showing the most potent effects, particularly against *S. aureus*. These results suggest that cassia leaf oil could be a valuable source of antimicrobial compounds. These findings prove that *C. cassia* leaf EO could be a valuable source of antioxidants and antibacterial compounds for various applications. This study supports the potential application and rational utilization of *C. cassia* leaves in the development of nutraceutical and functional food materials.

Temperature is an important environmental factor. However, the mechanism between this change and temperature or other environmental factors remains uncertain. Future studies should focus on how environmental factors affect the bioactivity accumulation of secondary metabolites in cinnamon plants and which regulatory genes or transcription factors affect the synthesis and accumulation of the main component, *trans*-cinnamaldehyde. The accumulation mechanism of important metabolites is revealed and applied to plant synthetic biology, which will expand the scope of application of plant secondary metabolites in the food and drug industry.

## Figures and Tables

**Figure 1 plants-14-00081-f001:**
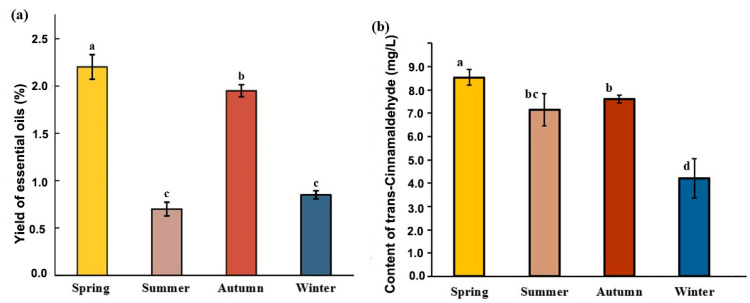
Yields (% *w*/*w*) (**a**) and *trans*-cinnamaldehyde (mg/L) (**b**) of *C. cassia* leaf essential oil from different harvested seasons. The columns with different lowercase letters are significantly different (*p* < 0.05).

**Figure 2 plants-14-00081-f002:**
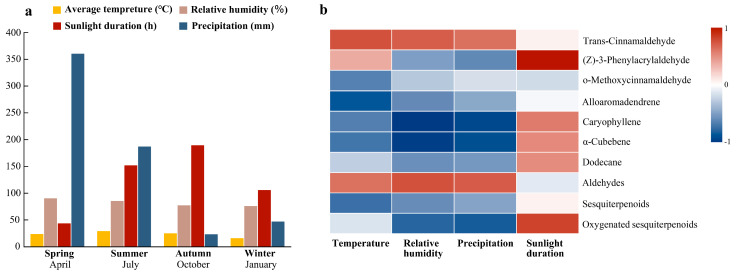
(**a**) Mean monthly weather parameters during the experimental period; (**b**) correlation between essential oil constituents of *C. cassia* and weather parameters from April 2019 to January 2020.

**Figure 3 plants-14-00081-f003:**
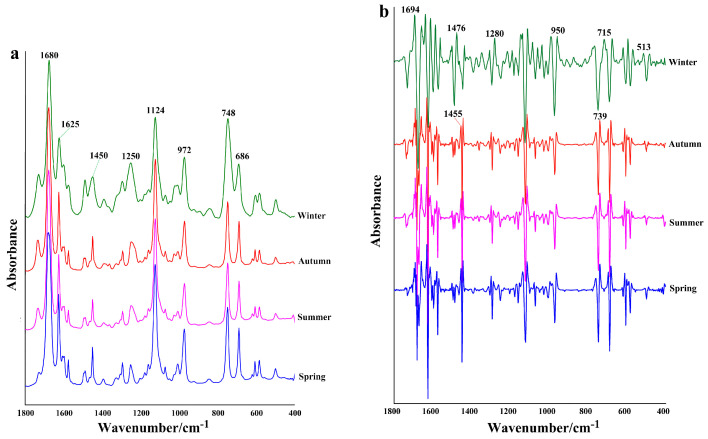
(**a**) IR spectra of leaf essential oils; (**b**) second derivative spectra of essential oils from the four samples.

**Figure 4 plants-14-00081-f004:**
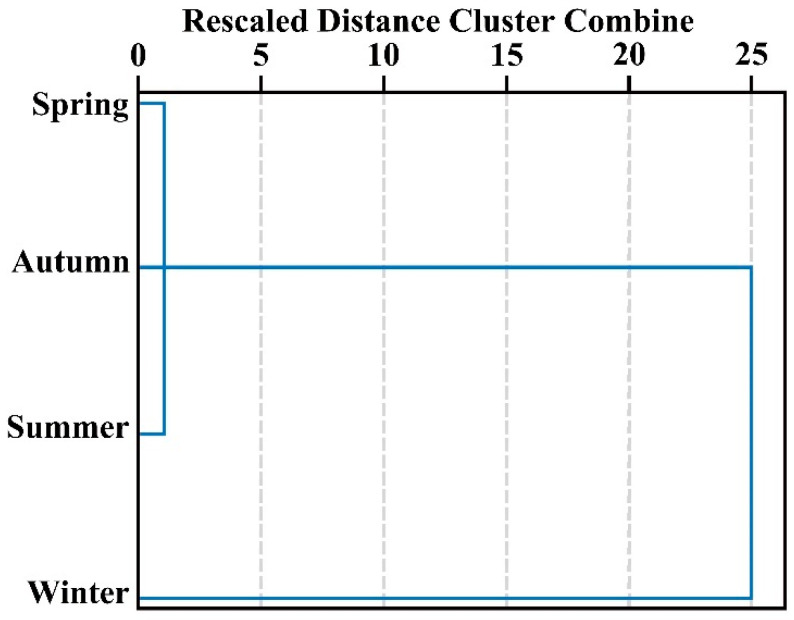
Cluster dendrogram constructed from the IR spectra for four samples.

**Figure 5 plants-14-00081-f005:**
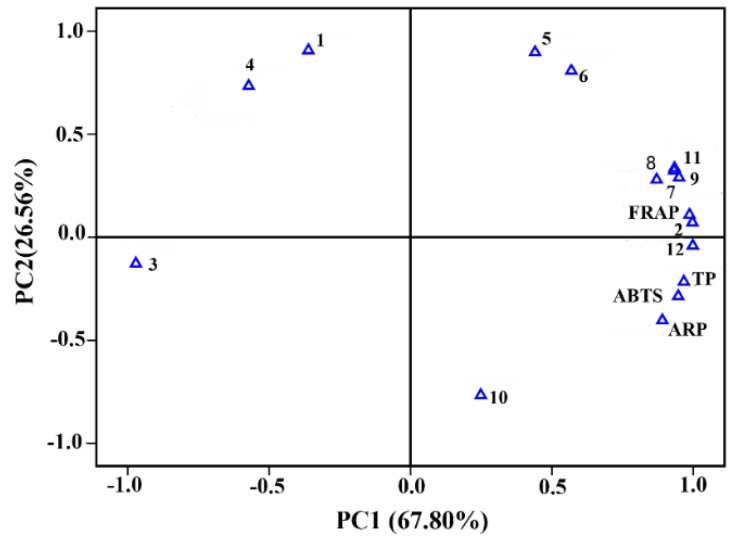
Principal component analysis loading plot of antioxidant activity and chemical compositions of EOs from *C. cassia* in different seasons. The numbers 1 to 12 and TP represent dodecane, *cis*-cinnamaldehyde, *trans*-cinnamaldehyde, (Z)-3-phenylacrylaldehyde, α-cubebene, caryophyllene, humulene, alloaromadendrene, γ-muurolene, α-muurolene, *cis*-bisabolene, o-methoxycinnamaldehyde, and total phenolic content, respectively.

**Figure 6 plants-14-00081-f006:**
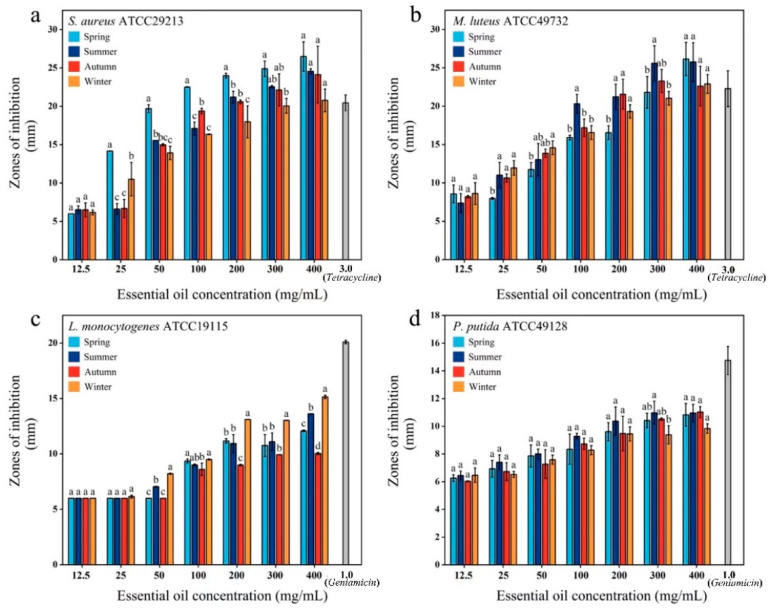
Antibacterial effect of cassia leaf EOs on *S. aureus* (**a**), *M. luteus* (**b**), *L. monocytogenes* (**c**), and *P. putida* (**d**) in different seasons. Least sensitive for a 6.00–10.0 cm diameter (+), moderately sensitive for a 10.00–20.00 cm diameter (++), and highly sensitive for a diameter greater than 20.00 cm (+++). The columns with different lowercase letters are significantly different (*p* < 0.05).

**Table 1 plants-14-00081-t001:** Main leaf EO components of *C. cassia* of the different seasons.

No.	Compound	RI	Relative Content%				Identification
			Spring	Summer	Autumn	Winter	
1	Benzenepropanal	1162	-	0.18 ± 0.01	-	-	GC–MS, RI
2	Dodecane	1177	1.65 ± 0.06	1.00 ± 0.07	2.73 ± 0.01	1.87 ± 0.07	GC–MS, RI
3	2-Methoxybenzaldehyde	1204	-	0.54 ± 0.01	0.20 ± 0.04	0.53 ± 0.02	GC–MS, RI
4	*cis*-Cinnamaldehyde	1215	-	0.46 ± 0.02	-	1.08 ± 0.01	GC–MS, RI
5	*trans*-Cinnamaldehyde	1227	92.59 ± 1.25	83.01 ± 1.56	91.10 ± 1.67	73.24 ± 1.78	GC–MS, RI, Co
6	(Z)-3-Phenylacrylaldehyde	1265	0.24 ± 0.02	0.26 ± 0.04	0.36 ± 0.04	0.25 ± 0.08	GC–MS, RI
7	α-Cubebene	1296	0.37 ± 0.04	0.41 ± 0.04	0.84 ± 0.07	0.99 ± 0.07	GC–MS, RI
8	Caryophyllene	1327	0.33 ± 0.03	0.41 ± 0.01	0.54 ± 0.03	0.64 ± 0.03	GC–MS, RI
9	Aromandendrene	1386	-	0.32 ± 0.01	-	0.10 ± 0.02	GC–MS, RI
10	Humulene	1454	-	0.18 ± 0.06	-	1.28 ± 0.05	GC–MS, RI
11	Alloaromadendrene	1465	0.18 ± 0.08	0.31 ± 0.02	0.22 ± 0.01	1.26 ± 0.04	GC–MS, RI
12	γ-Muurolene	1471	-	0.86 ± 0.03	0.47 ± 0.05	1.38 ± 0.06	GC–MS, RI
13	α-Bisabolol	1445	tr	0.18 ± 0.04	-	-	GC–MS, RI
14	α-Muurolene	1493	0.39 ± 0.01	1.16 ± 0.08	0.10 ± 0.01	0.38 ± 0.04	GC–MS, RI
15	*cis*-Bisabolene	1504	-	0.36 ± 0.07	tr	4.36 ± 0.58	GC–MS, RI
16	o-Methoxycinnamaldehyde	1522	3.4 ± 0.09	4.90 ± 0.54	2.08 ± 0.97	8.07 ± 0.54	GC–MS, RI
17	Benzylidenemalonaldehyde	1896	-	-	-	0.31 ± 0.01	GC–MS, RI
18	1R,3Z,9s-4,11,11-Trimethyl-8-methylenebicyclo [7.2.0]undec-3-ene	2042	tr	0.29 ± 0.03	tr	tr	GC–MS, RI
	Aldehydes		96.23	88.17	93.54	83.48	
	Sesquiterpenoids		0.94	3.6	1.63	9.75	
	Oxygenated sesquiterpenoids		0.33	0.59	0.54	0.64	
	Others		1.65	1.00	2.73	1.87	
Total identified			99.15	93.36	98.44	95.74	

RI: retention index; Co: co-injection with authentic compounds; -: not detected; tr (trace): relative content < 0.1%.

**Table 2 plants-14-00081-t002:** Correlation between essential oil constituents of *C. cassia* during April 2019–January 2020.

Chemical Constituent	2-Methoxybenzaldehyde	*trans*-Cinnamaldehyde	(Z)-3-Phenylacrylaldehyde	α-Cubebene	Caryophyllene	Alloaromadendrene	α-Muurolene
Dodecane	−0.877 *	0.449	0.820 *	0.716 *	0.566 *	−0.075	−0.966 *
2-Methoxybenzaldehyde		−0.823 *	−0.994 *	−0.293	−0.101	0.545 *	0.722 *
*trans*-Cinnamaldehyde			0.880 *	−0.301	−0.482	−0.924 *	−0.202
(Z)-3-Phenylacrylaldehyde				0.188	−0.007	−0.632 *	−0.643 *
α-Cubebene					0.981 *	0.642 *	−0.873 *
Caryophyllene						0.779 *	−0.761 *
Alloaromadendrene							−0.187

* Significant (*p* = 0.05).

**Table 3 plants-14-00081-t003:** Antioxidant activity of EOs from *C. cassia* in different seasons.

Season	Total Phenolic (mg GAE/g DW)	DPPH IC_50_ (μg/mL)	ARP	FRAP μmol Fe^2+^/g	ABTS μmol Trolox/g
Spring	19.66 ± 0.05 ^b^	21.55 ± 1.11 ^b^	0.04	19.66 ± 0.99 ^c^	12.21 ± 0.88 ^c^
Summer	23.82 ± 0.04 ^a^	20.37 ± 2.12 ^b^	0.049	24.75 ± 0.59 ^b^	13.67 ± 0.48 ^b^
Autumn	17.22 ± 0.07 ^b^	44.46 ± 1.92 ^a^	0.022	17.49 ± 0.29 ^d^	7.15 ± 0.41 ^d^
Winter	26.71 ± 0.06 ^a^	15.87 ± 3.23 ^c^	0.063	35.94 ± 0.39 ^a^	18.05 ± 0.06 ^a^

Values in the same column with different subscripts are significantly different for each assay at *p* < 0.05. GAE: gallic acid; DPPH: 1,1-Diphenyl-2-picrylhydrazyl; ARP: antiradical power; FRAP: ferric reducing/antioxidant power; ABTS: 2,2-azino-bis(3-ethylbenzothiazoline-6-sulphonic acid).

**Table 4 plants-14-00081-t004:** Minimum inhibitory concentration (MIC) and minimum bactericidal concentration (MBC).

	Season	*S. aureus* (mg/mL)	*M. luteus*	*L. monocytogenes* (mg/mL)	*P. putida* (mg/mL)
MIC (mg/mL^−1^)	Spring	0.25	0.50	0.25	1.00
	Summer	0.50	0.50	0.50	1.00
	Autumn	0.50	0.50	0.50	1.00
	Winter	0.50	0.50	0.50	1.00
MBC (mg/mL^−1^)	Spring	0.50	2.00	2.00	1.00
	Summer	1.00	2.00	1.00	1.00
	Autumn	0.50	2.00	2.00	1.00
	Winter	0.50	1.00	1.00	1.00

## Data Availability

Data are contained within the article.

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
