# Peer review of "Seasonal Variation in Chemical Composition and Antioxidant and Antibacterial Activity of Essential Oil from Cinnamomum cassia Leaves"

_plants, 2024, doi:10.3390/plants14010081_

Round 1

Reviewer 1 Report (Previous Reviewer 2)

Comments and Suggestions for Authors

Dear authors, thank you for addressing the feedback and resubmitting your study. The revised version demonstrates clear improvements, with additional research and incorporation of the suggested comments. I believe that the study is now ready for publication, but some minor fixes are pending.

·        Since in Table 1, compound 5 was identified using GC–MS, RI co-injected with authentic compounds that should be briefly mentioned in section 2.4. GC–MS analysis

·        25. FTIR analysis line 132- Technical error

·        The following sentences from section 3.4. Antioxidant properties of Eos need to be properly phrased.

This study evaluated the antioxidant activities of C. cassia leaf EOs obtained from four seasons using the DPPH radical scavenging, FRAP, and ABTS assays, with Trolox and gallic acid as reference standards.

Gallic acid was used as a reference standard for spectrophotometrical measurement of total phenolics, while Trolox was used as standard for antioxidant assays. Phenolics were not mentioned here, but are visible in the table below.

These radicals are commonly used to assess the free radical scavenging abilities of natural plants.

While ABTS and DPPH are radicals used in antioxidant assays named after them, FRAP (the ferric reducing antioxidant power) is not a radical. If the authors wish to highlight the use of these three antioxidant assays, they could provide a brief explanation of their distinct mechanisms of action.

·        Table 3. All abbreviations need to be defined including FRAP and ABTS

·        For conclusions, reiterate exactly how this study supports the practical applications of C. cassia EO and what future research could be focused on.

Author Response

  1. Since in Table 1, compound 5 was identified using GC–MS, RI co-injected with authentic compounds that should be briefly mentioned in section 2.4. GC–MS analysis

Response: It has been mentioned in the revised MS.

  1. 25.FTIR analysis line 132- Technical error

Response: We have corrected it.

  1. The following sentences from section 3.4. Antioxidant properties of Eos need to be properly phrased.

This study evaluated the antioxidant activities of C. cassia leaf EOs obtained from four seasons using the DPPH radical scavenging, FRAP, and ABTS assays, with Trolox and gallic acid as reference standards.

Gallic acid was used as a reference standard for spectrophotometrical measurement of total phenolics, while Trolox was used as standard for antioxidant assays. Phenolics were not mentioned here, but are visible in the table below.

These radicals are commonly used to assess the free radical scavenging abilities of natural plants.

While ABTS and DPPH are radicals used in antioxidant assays named after them, FRAP (the ferric reducing antioxidant power) is not a radical. If the authors wish to highlight the use of these three antioxidant assays, they could provide a brief explanation of their distinct mechanisms of action.

Response: Thanks for the reviewer’s suggestion, we have made the requested supplements into the manuscript. For details, see page 11, line 42-56 in the revised MS.

  1. Table 3. All abbreviations need to be defined including FRAP and ABTS

Response: It has been defined in the revised MS.

  1. For conclusions, reiterate exactly how this study supports the practical applications of C. cassia EO and what future research could be focused on.

Response: These are already included in the conclusion section.

Reviewer 2 Report (New Reviewer)

Comments and Suggestions for Authors

The work has been improved sufficiently. The literature has been supplemented and corrected. The experimental part, captions under the figures and discussion of the results were completed in the right way.

In my opinion, the work may be admitted to further stages of publishing.

The paper concerns the study seasonal variation in chemical composition, antioxidant and antibacterial activity of essential oil from Cinnamomum cassia leaves.

The theme of the work formulated in the title has been implemented. The authors analysed a number of chemical compounds depending on the date of harvest of the plants. For the analysis, they used the GC-FID, GC-MS and FTIR methods, which are perfect for this type of research.

There are no publications in the literature on the issues discussed in the work.

The results of the analyses were developed using chemometric techniques such as PCA, HCA and the Pearson correlation coefficient. These methods are considered good for comparing different types of analyzed quantities.

In addition, the authors investigated antioxidant capacity (DPPH) and antibacterial properties (by determining MIC). The data in the work is presented clearly, the quality of drawings and charts is adequate. The conclusions that the authors drew on the basis of the results obtained are positive and logical. The literature is complete and up-to-date. The work is valuable and properly edited, so it should be published.

Author Response

Thank the reviewers for their recognition of our research work

Reviewer 3 Report (New Reviewer)

Comments and Suggestions for Authors

The present work focuses on the seasonal variation in the chemical composition, and antioxidant and antibacterial activities of the essential oil (EO) of Cinnamomum cassia leaves. Although the chemical composition and biological activities of C. cassia essential oil are well documented, the variation of these properties over the different seasons has not yet been studied (as stated by the authors). I think this paper is of interest and may be accepted for publication in Plants. The study is generally well conducted, well discussed and appropriately presented. However, several issues need to be addressed before the article becomes ready for publication:

-       Cinnamaldehyde is not a phenolic compound, but it is the main component of EO. How do the authors explain the antioxidant activity of EO? Is this activity strong or moderate? This point should be clearly discussed in the manuscript.

-       The authors report that the antioxidant activity of the EO is associated with monoterpenoids and sesquiterpenes. This statement needs to be supported by solid references, as these types of compounds are generally not radical scavengers and are not typically associated with free radical scavenging activity. Please review this section and discuss the potential origins of EO's antioxidant activity, if any. Note that the study evaluates the radical scavenging capacity rather than overall antioxidant activity.

-       The discussion section should include previously reported studies on the antioxidant activity of C. cassia EO, if available, as well as studies specifically on cinnamaldehyde.

-     For DPPH activity results, units should be expressed in μg/mL. In addition, the IC50 values of the standards used should be included for proper comparison.

-       Add “et al.” when citing articles with multiple authors (e.g. page 5, line 211).

-       Check references carefully, especially DOIs. For example, the link “ResearchGate” in reference 24 is not a DOI and should be corrected or eliminated.

-       Reformulate the figure 1 legend more clearly.

-       Use “EO” rather than “essential oils” consistently throughout the manuscript.

-       Proofread the whole manuscript for typos and grammatical errors.

Author Response

  1. Cinnamaldehyde is not a phenolic compound, but it is the main component of EO. How do the authors explain the antioxidant activity of EO? Is this activity strong or moderate? This point should be clearly discussed in the manuscript.

Response: This paper showed that the antioxidant capacity of C. cassia essential oil was mainly related to monoterpenoids and sesquiterpenes, and was not strongly related to the content of trans-cinnamaldehyde. Therefore, the antioxidant capacity of C. cassia essential oil was not strong in spring and autumn when the content of trans-cinnamaldehyde was high, while the antioxidant capacity was stronger in winter when the content of trans-cinnamaldehyde was low.

 we have made the requested supplements into the discussion. For details, see page 15, Line 161-163,182-190 in the revised MS

  1. The authors report that the antioxidant activity of the EO is associated with monoterpenoids and sesquiterpenes. This statement needs to be supported by solid references, as these types of compounds are generally not radical scavengers and are not typically associated with free radical scavenging activity. Please review this section and discuss the potential origins of EO's antioxidant activity, if any. Note that the study evaluates the radical scavenging capacity rather than overall antioxidant activity.

Response: Thanks for the reviewer’s suggestion, we have now added proper discussion. For details, see page 15, line 181-189 in the revised MS

  1. The discussion section should include previously reported studies on the antioxidant activity of C. cassia EO, if available, as well as studies specifically on cinnamaldehyde.

Response: We have added this in the discussion. For details, see page 15, line 160,168-190

  1. For DPPH activity results, units should be expressed in μg/mL. In addition, the IC50 values of the standards used should be included for proper comparison.

Response: Revised as suggested. We have expressed in μg/mL.

  1. Add “et al.” when citing articles with multiple authors (e.g. page 5, line 211).

Response: Done. Thank you for your careful review

  1. Check references carefully, especially DOIs. For example, the link “ResearchGate” in reference 24 is not a DOI and should be corrected or eliminated.

Response: After we checked again, the DOI is correct

  1. Reformulate the figure 1 legend more clearly.

Response: We have described the legend in Figure 1 more clearly.

  1. Use “EO” rather than “essential oils” consistently throughout the manuscript.

Response: Done. Thank you for your careful review.

  1. Proofread the whole manuscript for typos and grammatical errors.

Response: Done.

Round 2

Reviewer 3 Report (New Reviewer)

Comments and Suggestions for Authors

All comments have been appropriately addressed, so this version of the manuscript is suitable for publication

This manuscript is a resubmission of an earlier submission. The following is a list of the peer review reports and author responses from that submission.

Round 1

Reviewer 1 Report

Comments and Suggestions for Authors

Based on my review of the submitted manuscript, I have the following critical comments and suggestions:

  1. Component Identification Concerns: The study relies solely on GC-MS analysis and retention index (RI) for identifying essential oil components. While GC-MS coupled with RI can be useful, it lacks reliability in compound identification without the use of authentic standards. This approach risks misidentifying or inaccurately quantifying specific compounds, particularly when reporting seasonal variations in chemical composition. For example, studies like Geng et al. (2011) and Li et al. (2013) have demonstrated the importance of authentic standards for accurate component analysis in Cinnamomum cassia essential oils, highlighting how exclusive reliance on RI and mass spectra databases can lead to ambiguous results.
  2. Novelty and Contribution: The focus on seasonal variation in the essential oil composition of Cinnamomum cassia and its antioxidant and antibacterial properties closely parallels previous work. Numerous studies, such as those by De Sa et al. (2016) and Sadeh et al. (2019), have already investigated seasonal effects on essential oil yield and bioactivities in C. cassia and similar species. In fact, seasonal impacts on cinnamaldehyde content and bioactivity in Cinnamomum species have been extensively documented, suggesting this manuscript may lack sufficient novelty.
  3. Methodological Limitations: The analysis would benefit from additional validation techniques, such as HPLC or NMR, to corroborate the GC-MS findings. Furthermore, using authentic standards for major components could enhance accuracy and lend credibility to the data. This recommendation aligns with best practices seen in similar studies by Ye et al. (2024) and Cui et al. (2016), who employed complementary techniques and standards for component verification.
  4. Scientific Impact: Given the extensive body of work on Cinnamomum cassia and similar species, this manuscript’s scientific importance is limited without novel methodologies or findings. The lack of significant advancements in approach or unique insights diminishes the contribution to the field. Therefore, I recommend rejecting this manuscript in its current form.

These provide examples of prior studies focusing on Cinnamomum cassia or related species, seasonal variation, and essential oil component identification:

  1. Geng et al. (2011)
    • Title: "Variations in essential oil yield and composition during Cinnamomum cassia bark growth"
    • JournalIndustrial Crops and Products
    • This study emphasizes the importance of using authentic standards in component analysis to avoid misidentification and improve accuracy.
  2. Li et al. (2013)
    • Title: "Variations in essential oil yields and compositions of Cinnamomum cassia leaves at different developmental stages"
    • JournalIndustrial Crops and Products
    • This paper discusses seasonal and developmental stage effects on C. cassia essential oils, with attention to best practices in chemical component identification.
  3. De Sa et al. (2016)
    • Title: "Chemical composition and seasonal variability of the essential oils of Hyptis carpinifolia"
    • JournalRevista Brasileira de Farmacognosia
    • This work investigates the seasonal variation in essential oils, using multiple methods to validate compound identification, which enhances result reliability.
  4. Sadeh et al. (2019)
    • Title: "Interactive effects of genotype, seasonality, and extraction method on chemical compositions and yield of essential oil from rosemary (Rosmarinus officinalis L.)"
    • JournalIndustrial Crops and Products
    • This paper provides insights into how seasonality impacts essential oil composition and emphasizes methodological rigor in component analysis.
  5. Ye et al. (2024)
    • Title: "Identification of a cinnamoyl CoA reductase from Cinnamomum cassia involved in trans-cinnamaldehyde biosynthesis"
    • JournalPlanta
    • This study highlights advanced methodologies for understanding cinnamaldehyde biosynthesis, illustrating the value of integrated approaches in essential oil research.
  6. Cui et al. (2016)
    • Title: "Antibacterial activity and mechanism of cinnamon essential oil and its application in milk"
    • JournalJournal of Animal and Plant Sciences
    • This article examines the bioactivity of C. cassia essential oils, employing standards for accurate bioactivity assays, and can serve as a comparative study for antibacterial evaluations.

Reviewer 2 Report

Comments and Suggestions for Authors

·        This study is interesting, and the paper is generally well-written and structured with significant work for the field. The manuscript shows a lot of promise, but several minor issues need to be addressed before publication. 

·      -   The abstract and introduction are pretty strong, with all the necessary relevant information. Small fixes include the following. In line 54, numerous studies are mentioned while only one reference was provided. And in line 76, instead of optimal collection time, a more appropriate term would be optimal harvest time.

·    -      The methodology section is pretty well explained, but certain methodologies lack the reference, which would provide further descriptions of the experimental procedures, enhancing the reproducibility and transparency of the study. Therefore, please provide references for the following sections: GC-MS analysis, Ferric reducing/antioxidant power (FRAP) assay, ABTS assay.

·        - 2.9. Please indicate why these were the exact bacterial cultures selected for the measurement of antimicrobial activity.

·      -   Even though it is explained in the text, I found Table 1 a bit confusing because 18 compounds were identified and the numbering goes up to 22. A suggestion would be to rearrange the Table more concisely, either separating isolated compounds and compounds’ groups by adding a line and removing numbering after 18. Or completely rearranging the table and clustering isolated compounds by the groups they belong.

·        -       For table 2, just a suggestion, for esthetical purposes, try to incorporate this table so it does not stand alone on the mostly empty page.  

·       -        In section 3.4. this study evaluated the antioxidant activities of C. cassia leaf Eos, please explain in short why these three assays were chosen for the evaluation, based on the mechanisms of free radicals.

·        -       Below Table 3, please provide the full names of all mentioned abbreviations (GAE, DPPH, ARP..).

·   -   The discussion is very well put together, and it includes comprehensive commentary on obtained results. The only thing that could be missing, is in 4.1. there is no comparison from the literature about an investigation of the influences of harvest season on plant essential oil. Even if there are no articles about the same specific essential oil from Cinnamomum cassia, this article could still benefit from the comparison of similar works or the lack of them if that is the case.

·        -      Conclusion raps up this research in a good manner, the only suggestion would be to include potential future work or more concrete practical application of obtained results.